# Genome-Wide Identification and Characterization of the Phytochrome Gene Family in Peanut

**DOI:** 10.3390/genes14071478

**Published:** 2023-07-20

**Authors:** Yue Shen, Yonghui Liu, Man Liang, Xuyao Zhang, Zhide Chen, Yi Shen

**Affiliations:** Institute of Industrial Crops, Jiangsu Academy of Agricultural Sciences, Nanjing 210014, China; syjaas@163.com (Y.S.);

**Keywords:** phytochrome, *Arachis hypogaea*, genome-wide characterization, gene family, bioinformatics analysis

## Abstract

To investigate the potential role of phytochrome (PHY) in peanut growth and its response to environmental fluctuations, eight candidate *AhPHY* genes were identified via genome-wide analysis of cultivated peanut. These AhPHY polypeptides were determined to possess acidic and hydrophilic physiochemical properties and exhibit subcellular localization patterns consistent with residence in the nucleus and cytoplasm. Phylogenetic analysis revealed that the *AhPHY* gene family members were classified into three subgroups homologous to the *PHYA/B/E* progenitors of *Arabidopsis*. *AhPHY* genes within the same clade largely displayed analogous gene structure, conserved motifs, and phosphorylation sites. *AhPHY* exhibited symmetrical distribution across peanut chromosomes, with 7 intraspecific syntenic gene pairs in peanut, as well as 4 and 20 interspecific *PHY* syntenic gene pairs in *Arabidopsis* and soybean, respectively. A total of 42 *cis*-elements were predicted in *AhPHY* promoters, including elements implicated in phytohormone regulation, stress induction, physiology, and photoresponse, suggesting putative fundamental roles across diverse biological processes. Moreover, spatiotemporal transcript profiling of *AhPHY* genes in various peanut tissues revealed distinct expression patterns for each member, alluding to putative functional specialization. This study contributes novel insights into the classification, structure, molecular evolution, and expression profiles of the peanut phytochrome gene family, and also provides phototransduction gene resources for further mechanistic characterization.

## 1. Introduction

Light constitutes a crucial environmental factor that regulates plant growth and developmental dynamics, governing processes such as photosynthesis and photomorphogenetics. Previous studies have demonstrated that plant photochromic systems predominantly consist of three classic families of photoreceptors: phytochromes (PHY) that detect red/far-red (R/FR) light, cryptochromes (CRY), and phototropins (PHOT) that detect blue/UV-A light. These photosensory chromoproteins are capable of accurately perceiving various attributes of ambient light including wavelength, intensity, directionality, and periodicity [1].

Phytochromes are capable of photo-interconversion between the biologically active far-red light-absorbing form (Pfr) and the inactive red light-absorbing form (Pr) upon exposure to red light and far-red light, respectively [2,3]. The typical structure of plant phytochromes consists of a highly conserved N-terminal PAS-GAF-PHY* photosensory module and a C-terminal HKRD regulatory module. A cysteine residue on the GAF domain covalently anchors a tetrapyrrole chromophore, whose inherent photochemical properties govern the reversible photoisomerization of the phytochrome between the Pr and Pfr conformers [4,5].

The model plant *Arabidopsis thaliana* possesses five phytochrome genes (*PHYA-PHYE*) that have evolved through a series of duplication events [6]. While sharing some degree of sequence similarity, these phytochromes display divergent biochemical properties and significant functional diversity during plant morphogenesis. Studies have shown that PHYA exhibits rapid Pfr instability and can transduce signals upon rapid photoreversible conversion between Pr and Pfr forms, thus promoting plant de-etiolation. PHYA specifically mediates the very-low-fluence response (VLFR) regulating seed germination under very low light and the far-red high-irradiance response (FR-HIR) regulating seedling morphogenesis under vegetation shade. In contrast, PHYB/C/D/E are relatively Pfr-stable, with PHYB functioning as the primary photosensor mediating the classic reversible red/far-red low-fluence response (LFR) and red high-irradiance response (R-HIR) that govern nearly all stages of plant development [7,8,9].

Phytochromes have been extensively characterized in multiple crop species such as rice [10], millet [11], maize [12,13], mungbean [14], soybean [15,16], potato [17] and tomato [18,19] utilizing genetic and molecular techniques. However, the phytochrome gene family of peanut remains poorly characterized. By leveraging the genome data of cultivated peanut in conjunction with bioinformatics tools, this study identified and analyzed the physiochemical properties, structural features, molecular evolution, and putative functions of peanut phytochromes, providing a theoretical framework for in-depth mechanistic characterization of the *AhPHY* gene family.

## 2. Materials and Methods

### 2.1. Prediction of PHY Genes in Peanut

The experimental material for this study was the cultivated peanut (*Arachis hypogaea* L. cv. Tifrunner). Genome and protein sequences for cultivated peanut were obtained from the Peanutbase database (https://www.peanutbase.org/, accessed on 15 January 2023) and locally compiled. Based on the conserved domains of the phytochrome gene family in *Arabidopsis*, hidden Markov models of the PHY* (PF00360) and GAF (PF01590) domains were retrieved from the InterPro database (https://www.ebi.ac.uk/interpro/, accessed on 14 January 2023). The PHY*.OUT and GAF.OUT seed alignments were constructed using the hmmbuild of the HMMER tool (version 3.0) and hmmsearch of CMD locally. The polypeptide sequences were then extracted using SeqHunter tool (version 1.0) with an E-value cutoff of 1 × 10^−10^. Alignments of preselected protein sequences were performed with the SMART sequence analysis tool (http://smart.embl-heidelberg.de/, accessed on 15 January 2023). Repetitive sequences and redundant transcripts were eliminated, and the candidate members of the peanut *PHY* gene family were identified. 

Multiple sequence alignment analysis was performed on the *PHY* family protein sequences of peanut (*Arachis hypogaea*), soybean (*Glycine max*), and *Arabidopsis* (*Arabidopsis thaliana*) using Clustal X software (version 1.83) [20]. A phylogenetic tree was constructed using the maximum likelihood method in MEGA X software (version 10.1.8) with a bootstrap value of 1000 [21]. Data on *Arabidopsis* and soybean were obtained from the TAIR database (https://www.arabidopsis.org/, accessed on 5 February 2023) and Phytozome database (https://Phytozome-next.jgi.doe.gov/, accessed on 5 February 2023), respectively.

### 2.2. Sequence Structure Analysis

To characterize the gene features of *AhPHY*, various bioinformatics tools for in silico analyses were employed. The physicochemical properties were analyzed using the ProtParam tool from ExPASy (https://web.expasy.org/protparam/, accessed on 7 February 2023) to determine the number of amino acids (aa), molecular weight (MW), theoretical isoelectric points (pI), instability index (II), aliphatic index, and grand average of hydropathicity (GRAVY). The presence and location of signal peptides were predicted using SignalP version 5.0 (https://services.healthtech.dtu.dk/services/SignalP-5.0/, accessed on 8 February 2023). Transmembrane helices were predicted using TMHMM version 2.0 (https://services.healthtech.dtu.dk/services/TMHMM-2.0/, accessed on 8 February 2023). Subcellular localization was predicted using ProtComp version 9.0 (http://linux1.softberry.com/berry.phtml?topic=protcomppl&group=programs&subgroup=proloc/, accessed on 8 February 2023). Phosphorylation sites for serine, threonine, and tyrosine were predicted using NetPhos version 3.1 (https://services.healthtech.dtu.dk/service.php?NetPhos/, accessed on 12 February 2023).

Conserved motifs of candidate proteins were analyzed using the MEME suite version 5.5.3 (https://meme-suite.org/meme/tools/meme/, accessed on 9 February 2023) with the following parameters: maximum 10 motifs, minimum motif width 6, and maximum motif width 50 [22]. The resulting meme.xml file was downloaded for further analysis. To confirm the presence of conserved domains, the batch CD-search tool from NCBI (https://www.ncbi.nlm.nih.gov/cdd/, accessed on 10 February 2023) was utilized and the resulting hitdata.txt file was downloaded for further analysis. The conserved motifs and domains in candidate genes were both visualized using TBtools software (version 1.112) [23]. Coding (CDS) and genomic sequences (in FASTA format) for candidate genes were retrieved from local datasets, then gene structure visualization was performed using GSDS version 2.0 (http://gsds.gao-lab.org/, accessed on 10 February 2023) [24].

The secondary structures of candidate proteins were predicted using the SOPMA web server (https://npsa-prabi.ibcp.fr/cgi-bin/npsa_automat.pl?page=/NPSA/npsa_sopma.html/, accessed on 15 February 2023) with default parameters. While tertiary protein structures were modeled using the SWISS-MODEL web server (https://swissmodel.expasy.org/, accessed on 15 February 2023) with default parameters.

### 2.3. Chromosomal Localization and Syntenic Analysis

Chromosomal location information of the peanut *PHY* gene family was retrieved from local datasets, and physical distribution of candidate genes on peanut chromosomes was annotated using TBtools. Based on the genomic and structural annotation datasets of peanut, *Arabidopsis*, and soybean, collinearity maps of peanut *PHY* genes within and across species were constructed and visualized using TBtools, then exported as an SVG vector graphic file, which was further refined using Adobe Illustrator software (version 15.0.0).

### 2.4. Cis-Elements Prediction and Expression Pattern Analysis

The 2000 bp upstream promoter regions proximal to the start codon of the CDS for each *AhPHY* gene were retrieved from the local cultivated peanut genome dataset using SeqHunter based on sequence coordinates. Putative *cis*-elements within candidate genes were then predicted using PlantCARE (http://bioinformatics.psb.ugent.be/webtools/plantcare/html/, accessed on 13 February 2023) [25] and complied into local datasets. The positions of predicted *cis*-elements were visualized using the GSDS tool and refined using Adobe Illustrator for final figure preparation. 

Transcriptome datasets from cultivated peanut and diploid progenitor were obtained from Peanutbase and locally compiled. Transcript abundance for candidate genes was quantified using the fragments per kilobase million (FPKM) metric, calculated with Featurecounts software (version 2.0.3) [26]. The tissue expression heatmaps were constructed and visualized using TBtools. FPKM values underwent log-transformation and row-normalization for visualization within the squared heatmap. Raw FPKM values were displayed without normalization in the circular heatmap. These heatmaps were combined and rendered in TBtools, then exported as an SVG vector graphic file, which was further refined using Adobe Illustrator for final figure preparation.

## 3. Results

### 3.1. Identification of the PHY Gene Family in Peanut

Based on the genome-wide data of cultivated peanut in Peanutbase, two hidden Markov models corresponding to the PHY* (PF00360) and GAF (PF01590) domains were utilized as queries to screen 9 and 18 polypeptide sequences harboring the aforementioned motifs, respectively. Through alignment of the aforementioned preliminary screened protein sequences using the SMART tool, 8 candidate members of the peanut phytochrome family were obtained and denoted as *AhPHY*. Protein physicochemical properties were analyzed using the ExPASy-ProtParam tool (Table 1). *AhPHY* encode proteins ranging from 1101 to 1151 amino acids in length, with molecular weights ranging from 122.42 to 128.16 kDa, and theoretical isoelectric points from 5.72 to 6.14, characterizing them as acidic proteins (pI < 7). The aliphatic index ranges from 91.04 to 95.50 and the instability index ranges from 43.27 to 46.13, characterizing them as unstable proteins (II > 40). The grand average hydropathy (GRAVY) index ranges from −0.055 to −0.171, indicating a degree of hydrophilicity. Furthermore, all AhPHY proteins lack an N-terminal signal peptide and transmembrane helices, indicating an intracellular localization. In silico subcellular localization prediction showed that AhPHY exhibits potential for multiple localizations, including the nucleus and cytoplasm.

Phosphorylation sites represent crucial protein functional loci that play an important role in regulating cellular function. According to phosphorylation site prediction by the NetPhos tool, the AhPHY proteins contain serine (Ser), threonine (Thr), and tyrosine (Tyr) residues, ranging from 96 to 104 across proteins. Specifically, Arahy.T2CQE4 encompasses the largest number of Ser residues (64), while Arahy.HS5Z9Z contains the fewest (57). Arahy.HS5Z9Z also exhibits the most Thr residues (37), whereas Arahy.PM8GQZ possesses the fewest (24). Additionally, Arahy.F3Y113 contains the most Tyr residues (11), while both Arahy.7E2TSQ and Arahy.D04KR2 harbor the fewest (8).

### 3.2. Phylogenetic Analysis and Classification of AhPHY Genes

To further elucidate the phylogeny and functional characteristics of the peanut *PHY* family, a cluster analysis of phytochrome gene homologs from peanut, *Arabidopsis*, and soybean was performed using the maximum likelihood method in the MEGA X software (Figure 1). Based on gene structure and sequence homology, the homologs were classified into five subgroups. Only members homologous to the *PHYA/B/E* clades were identified in the peanut and soybean genomes, with the most recent gene duplication event in this family occurring in the *PHYB* progenitor. Based on photoinduced activity, they can also be categorized into two types: type I exhibits photoactivation kinetics consistent with photolability and predominantly encompasses *PHYA*-like homologous sequence, while type II exhibits photostability and its sequence homology to *PHYB/C/D/E*.

### 3.3. Conserved Motif, Domain, and Structure Analysis

Gene structure analysis revealed that, with the exception of *Arahy.NNA8KD*, which contained 5 exons and 4 introns, the remaining *AhPHY* genes contained 4 exons and 3 introns, exhibiting a similar gene structure. Concomitantly, members of each subtype exhibited a degree of genetic homology, with phylogenetic nearness correlating positively with similarities in UTR length and distribution as well as exon number and organization, indicating conservation of gene structure among closely related proteins (Figure 2A,B). For a gene family, shared motifs may be directly associated with gene structure and function. Ten conserved motifs were identified using the MEME tool. Motifs 1-10 were present in all *AhPHY* in the same order. Motifs 10/2/5 and 6/4/8/1 exhibited the highest degree of conservation, constituting the recognition sites of the GAF and PHY* domains of phytochrome proteins, respectively. Additionally, motifs 7/3/9 constituted the recognition site of the PAS domain (Figure 2C). Domain analysis revealed that all AhPHY proteins harbored the characteristic COG4251 superfamily domain and PAS domain. COG4251 is classified as a model spanning multiple domains and constitutes the region of the histidine kinase responsible for photoinduced signal transduction, whereas the PAS domain exists as two tandem repeats and typically functions as a signal sensor for light and oxygen in signal transduction.

According to secondary structure prediction by the SOPMA tool, the AhPHY proteins were found to encompass α-helix, extended strand, β-turn, and random coil secondary structural elements. The α-helix content was the highest overall, constituting 47.35–49.28%, followed by random coil, constituting 30.34–32.80%, and extended strand, constituting 14.31–14.80%, while the β-turn was the lowest, constituting 4.89–5.81% (Table 2). Prediction using the SWISS-MODEL tool revealed that the tertiary structures of all AhPHY proteins were largely similar, especially between allelic variants (Figure 3), which was consistent with the aforementioned gene structure analysis outcomes. 

### 3.4. Genome Distribution and Syntenic Analysis of AhPHY Genes

The chromosomal localizations of the *AhPHY* genes were mapped using TBtools. As shown in Figure 4, the peanut genome (AABB) encompasses 20 chromosomes, with 8 candidate genes located on chromosomes Chr03, Chr04, Chr06, Chr09, Chr13, Chr14, Chr16, and Chr19, respectively, demonstrating an overall symmetrical distribution. Intraspecific collinearity analysis revealed seven pairs of genomic synteny between six genes of the peanut *PHY* gene family. Specifically, *Arahy.7E2TSQ* on Chr06, *Arahy.HS5Z9Z* on Chr09, *Arahy.D04KR2* on Chr16, and *Arahy.NNA8KD* on Chr19 were collinear with each other. *Arahy.A3SZXW* on Chr04 and *Arahy.T2CQE* on Chr14 were also collinear. Thus, it can be inferred that most *AhPHY* may have arisen through whole genome duplication events, indicating that segmental duplication played an important role in driving the evolution of the peanut phytochrome gene family.

To further elucidate the phylogenetic mechanisms underlying the peanut *PHY* gene family, we constructed a genome-wide collinearity map of peanut, *Arabidopsis*, and soybean, highlighting the *PHY* gene family members exhibiting collinear relationships across the three species. The results demonstrated that four peanut *AhPHY* genes and one *Arabidopsis AtPHY* gene formed four syntenic gene pairs, whereas six peanut *AhPHY* genes and six soybean *GmPHY* genes formed twenty syntenic gene pairs (Figure 5), indicating that the *PHY* family between peanut and soybean shares an exceptionally high degree of homology. These *PHY* syntenic gene pairs may possess similar potential functions or even originate from a common ancestor.

### 3.5. Promoter Cis-Element Analysis of AhPHY Genes

To explore the putative transcriptional regulatory mechanism of the peanut *PHY* gene family, *cis*-acting element prediction was performed on the upstream 2000 bp promoter sequences of each member. A plethora of *cis*-acting elements were enriched in the promoter regions of the peanut *PHY* gene family, including those responsive to plant hormones (auxin, gibberellin, abscisic acid, ethylene, methyl jasmonate, and salicylic acid), abiotic stresses (anaerobic induction, drought induction, osmotic stress, low temperature response, and defense response, etc.), growth and development (flavonoid biosynthesis, meristem expression, endosperm expression, and circadian rhythm, etc.), as well as various light-responsive elements (Figure 6A). 

Statistical analysis revealed that, with the exception of *Arahy.HS5Z9Z*, which lacked *cis*-elements associated with physiological function, the remaining *AhPHY* members harbored all four aforementioned modules of *cis*-elements. *Arahy.A3SZX* contained the most types of *cis*-elements (21 types) and *Arahy.NNA8KD* contained the fewest (12 types). Regarding phytohormone regulation, *AhPHY* contained three to five types of hormone response elements, and each member contained abscisic acid response elements (ABRE and AAGAA-motif) and methyl jasmonate response elements (CGTCA-motif and TGACG-motif). Four members contained auxin response elements (TGA-element and AuxRR-core), four members contained gibberellin response elements (GARE-motif and TATC-box), and all members except *Arahy.7E2TSQ* and *Arahy.PM8GQZ* contained ethylene response elements (ERE), whereas three members contained salicylic acid response elements (TCA-element and SARE). A total of twelve photoresponsive elements were identified in the *AhPHY* gene family, with one to five types distributed in each member. *Arahy.7E2TSQ* contained the most types of photoresponsive elements, whereas *Arahy.NNA8KD* contained the fewest types but the highest number. Regarding plant growth and development regulation, only *Arahy.A3SZXW* contained vascular bundle specific expression element (AC-I), *Arahy.7E2TSQ* contained flavonoid biosynthesis element (MBSI), *Arahy.T2CQE4* contained circadian rhythm control element (circadian), five members contained meristem expression regulatory element (CAT-box), and two members contained endosperm expression regulatory element (O2-site) (Figure 6B). 

### 3.6. Expression Pattern Analysis of AhPHY Genes

RNA-Seq read counts from 22 tissues of cultivated peanut were obtained from Peanutbase to analyze the spatiotemporal expression patterns of *AhPHY*. Members sharing closer phylogenetic relationships displayed similar spatiotemporal expression trends. As shown in Figure 7, the expression levels of *Arahy.HS5Z9Z* and *Arahy.NNA8KD* were the highest during early seed development, followed by pod development, and the lowest in leaves. The expression level of *Arahy.D04KR2* was the highest in nodules, followed by pistils, and it was also highly expressed during later seed development, whereas *Arahy.7E2TSQ* was predominantly expressed in leaves, pistils, and mid-developmental seeds. *Arahy.PM8GQZ* and *Arahy.F3Y113* were mainly expressed in leaves and shoots but barely expressed during seed development. *Arahy.T2CQE4* and *Arahy.A3SZXW* were mainly expressed in leaves, shoots, and early-developmental seeds.

Furthermore, the original expression levels in each tissue were compared based on their circular scale values, followed by an analysis of the tissue-specific expression patterns of *AhPHY*. The results demonstrated that *Arahy.HS5Z9Z* and *Arahy.NNA8KD* exhibited the highest expression abundance in gynophores, pods, as well as seeds during early development. *Arahy.D04KR2* showed the highest expression levels in nodules, pistils, and seeds during later development. *Arahy.T2CQE4* and *Arahy.A3SZXW* had the highest expression levels in leaves and shoots. These results imply that *AhPHY* may play a role in the growth and development of different tissues in peanut.

To study the expression profile of the peanut *PHY* gene family in response to abiotic stress, transcriptome sequencing analysis was performed on peanut gynophores subjected to prolonged darkness for 7 days. The analysis revealed that all *AhPHY* genes except *Arahy.F3Y113* were up-regulated to varying extents under dark treatment, with *Arahy.NNA8KD* exhibiting the highest basal expression level and showing the greatest up-regulation, representing an almost 73.2% increase in transcript abundance relative to the control (Appendix A). Additionally, transcriptome datasets of the diploid progenitor *Arachis duranensis* responding to drought stress and nematode infection were obtained from Peanutbase [27,28], and the expression patterns of *AhPHY* genes were analyzed (Appendix A). The results demonstrated that under drought treatment the transcripts homologous to *Arahy.7E2TSQ* were up-regulated while those homologous to *Arahy.NNA8KD*, *Arahy.PM8GQZ*, and *Arahy.A3SZXW* were down-regulated, exhibiting a similar trend as observed in drought-treated seedlings of cultivated peanut (Appendix A) [29]. Furthermore, following nematode infection for zero, three, six, and nine days, the abundance of *Arahy.7E2TSQ* homologous transcripts initially decreased and then increased whereas the other homologous transcripts exhibited the opposite pattern, indicating that the *AhPHY* genes may play an important role in abiotic and biotic stress responses.

## 4. Discussion

Phytochromes are implicated in light-induced development throughout plant ontogenesis, including seed germination, seedling photomorphogenesis, shade avoidance response, flowering induction, and senescence of adult plants, etc. [7]. Concomitantly, the evolution of distinct subgroups of photoreceptors within the phytochrome gene family enhanced plant sensitivity to fluctuating light qualities, conferring them with functional diversity [9,30] and promoting interaction with other endogenous signaling transduction pathways (e.g., plant hormones) [31,32,33,34,35,36].

Peanut is a widely cultivated oilseed and economic crop globally, with its yield and quality formation governed by both environmental and genetic factors. In this study, eight *AhPHY* members encoding 1101 to 1151 amino acids were identified in the genome-wide analysis of cultivated peanut, all of which belong to acidic hydrophilic proteins. Analysis of phytochrome subcellular localization in transgenic *Arabidopsis* seedlings expressing five genotypes of PHY:GFP fusion proteins revealed that under dark conditions, fluorescence was localized to the cytoplasm. Following light exposure, the fusion proteins translocated to the nucleus and aggregated into characteristic spots, exhibiting typical circadian rhythm regulation [37,38,39,40]. Subcellular localization prediction revealed that the *AhPHY* gene family members localize predominantly to the nucleus and cytoplasm, consistent with the nuclear-cytoplasmic distribution characteristics of phytochromes reported previously.

Phylogenetic analysis demonstrated that based on photosensitivity and light-induced characteristics, the peanut *PHY* gene family encompasses two physical types: photolabile type I and photostable type II [41]. Concomitantly, according to gene structure and sequence similarity, *AhPHY* can be further subdivided into three subgroups homologous to *PHYA/B/E* progenitors, containing four, two, and two homologous copies, respectively. This classification result is highly analogous to that of soybean [15]. Genetic variability depends on gene structure and conserved domains [42]. In this study, the gene structure of the *AhPHY* gene family was relatively conserved during evolution, with an identical number and distribution of conserved motifs. All members contain the typical COG4251 superfamily domain and PAS domain of phytochrome proteins, indicating a high degree of conservation in both evolution and function.

Protein phosphorylation constitutes an extensive and crucial post-translational modification process, closely associated with various biological processes, such as cellular signal transduction, DNA damage repair, transcriptional regulation, apoptosis regulation, etc. [43,44]. Predicting phosphorylation sites aids in elucidating protein phosphorylation modification mechanisms of target proteins. In this study, the number of phosphorylation sites of *AhPHY* members with close phylogenetic relationships was similar, as was the distribution of serine, threonine, and tyrosine residues. The *PHYA* and *PHYB* branches contained more phosphorylation sites, indicating that the member genes of these two branches may be more sensitive to changes in the external light environment and respond more rapidly.

Compared with the five classical clades of the *Arabidopsis PHY* gene family phylogenetic tree, the eight *AhPHY* genes identified here were distributed across only three ancestral clades. Chromosomal localization revealed a symmetrical distribution, indicating that each clade may have divergent or redundant gene copies. These findings imply that gene duplication contributed a pivotal role in the expansion and evolution of the allotetraploid peanut *AhPHY* gene family. Collinearity analysis revealed seven pairs of genomic synteny between six *AhPHY* genes within species of peanut. Furthermore, four *AhPHY* genes in peanut and one *AtPHY* gene in *Arabidopsis* formed four syntenic gene pairs, whereas six *AhPHY* genes in peanut and six *GmPHY* genes in soybean formed twenty syntenic gene pairs. The high degree of synteny between peanut and soybean *PHY* genes indicates a conservation of function that may reflect their closer evolutionary relationship as legumes. By contrast, the limited synteny between peanut and *Arabidopsis PHY* genes points to greater functional diversification since their last common ancestor. In summary, integrating phylogenetic and collinearity analyses may enable prediction of the putative biological functions of the peanut *PHY* gene family by comparing them with homologous *PHY* genes in model plants and related species.

A total of 42 *cis*-regulatory elements responsive to phytohormones, abiotic stresses, growth and development, as well as photoresponse, were identified within the putative promoter regions of genes encoding the peanut *PHYTOCHROME* photoreceptor protein family. The differences in the identity and combination of *cis*-elements between these photoreceptors reflect the diversity of their spatio-temporal expression patterns and putative participation in complex signaling networks. In our study, the *AhPHY* gene family was found to contain elements responsive to light, drought, and defense stress, wherein the majority of these members exhibited significant changes in expression levels following exposure of complete photodeprivation, osmotic stress, and nematode infection. These findings suggest that *AhPHY* genes may fulfill important functions related to plant photomorphogenesis [6,45] and tolerance to abiotic and biotic stresses [46,47,48]. Meanwhile, the independent presence of specific *cis*-elements inside the promoters of distinct *PHY* family members indicates their significant non-redundant functions. For example, the circadian regulatory element present within the promoter of the *Arahy.T2CQE4* gene, localized between −1977 bp and −1986 bp upstream of the transcription start site, implies a unique regulatory role for this gene in the plant’s circadian rhythm responses. The presence of this *cis*-element provides scientific evidence in support of the hypothesized function of the *Arahy.T2CQE4* gene. 

The spatiotemporal expression patterns of *AhPHY* gene family members were rather interesting. Members exhibiting close phylogenetic relationships demonstrated similar spatiotemporal expression patterns, whereas members of distinct clades exhibited certain tissue specificity. Accordingly, the putative relationship between gene expression and plant ontogeny can be inferred from spatiotemporal expression trends and expression abundance. Members exhibiting constitutive expression may play a role in the maintenance of basic cell and organ functions, while those with high expression during reproductive development are likely involved in light or photoperiod regulation of pod and seed formation. For instance, *Arahy.HS5Z9Z* and *Arahy.NNA8KD* may be related to the morphogenetic processes occurring early during the development of pods and seeds, and the organogenesis of floral organs. Whereas *Arahy.T2CQE4* and *Arahy.A3SZXW* may be associated with the seed embryogenesis, as well as photomorphogenesis of leaves within each growth period. *Arahy.D04KR2* may participate in processes of root nodulation and pistil differentiation. Furthermore, the transcripts level of *Arahy.PM8GQZ* and *Arahy.F3Y113* displayed limited variation across various tissues in comparison to other phytochrome family members. Their expression profiles were largely constitutive at a relatively low baseline level. This pattern implies that these genes may encode constitutively expressed proteins or exhibit redundancy function during plant ontogenesis [49].

## 5. Conclusions

In this study, eight members of the peanut phytochrome gene family were identified and characterized using bioinformatic approaches. Comprehensive analyses of their physicochemical properties, phylogenetic relationships, sequence structures, chromosomal distributions, collinearities, promoter *cis*-elements, and expression patterns provide a scientific framework for further elucidating the biological functions of *AhPHY* in peanut growth, development, and environmental adaptation.

## Figures and Tables

**Figure 1 genes-14-01478-f001:**
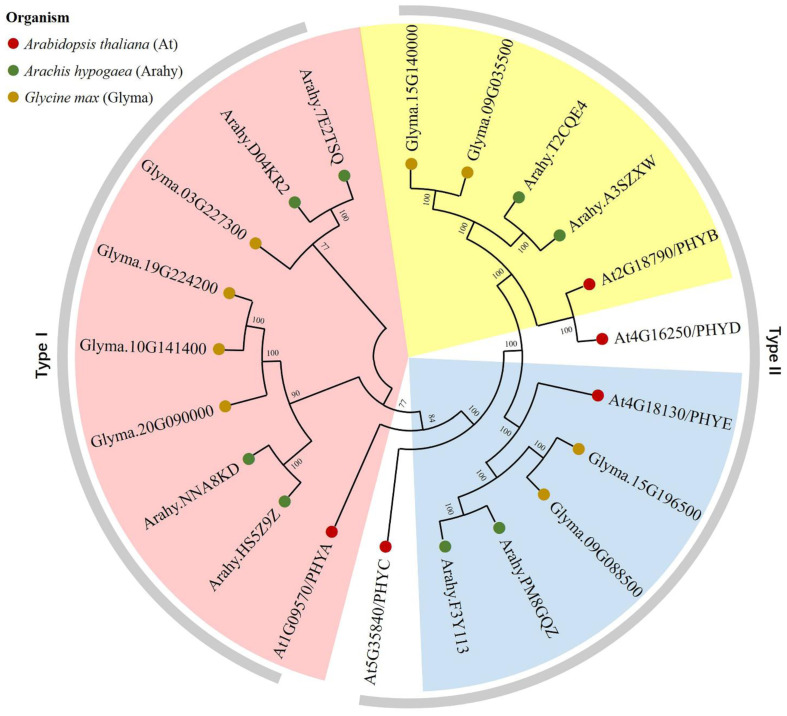
Phylogenetic analysis of *PHY* genes in peanut, soybean, and *Arabidopsis*. The *PHY* genes are denoted by green, khaki, and red dots for peanut, soybean, and *Arabidopsis*, respectively. The subgroups are delineated by colored fan-shaped partitions homologous to 5 *PHYs* (*PHYA-PHYE*) in *Arabidopsis*. In the outer ring (grey), the classification is established upon photoinduced activity, including photolabile type I and photostable type II variants.

**Figure 2 genes-14-01478-f002:**
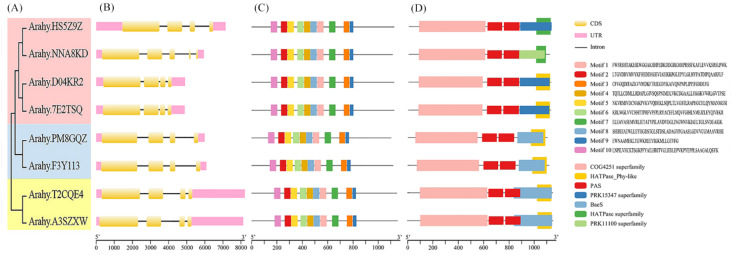
Phylogenetic tree, gene structure, conserved motifs, and domains of *PHY* genes in peanut. (**A**) A maximum likelihood phylogenetic tree of *AhPHY* genes was constructed using the MEGA X tool; (**B**) The exon-intron architecture of *AhPHY* genes was performed using the GSDS tool; (**C**) Motif composition of AhPHY proteins was visualized by TBtools, with different colored boxes representing motifs containing specific amino acid sequences; (**D**) Domain architecture of AhPHY proteins visualized by TBtools, with different colored boxes representing conserved domains.

**Figure 3 genes-14-01478-f003:**
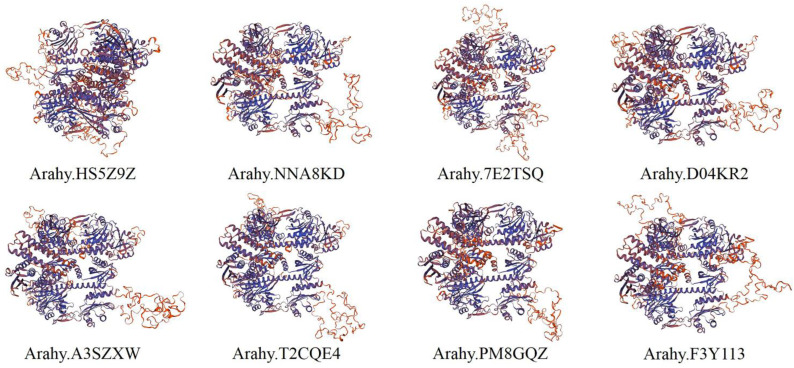
Tertiary structure model of AhPHY proteins.

**Figure 4 genes-14-01478-f004:**
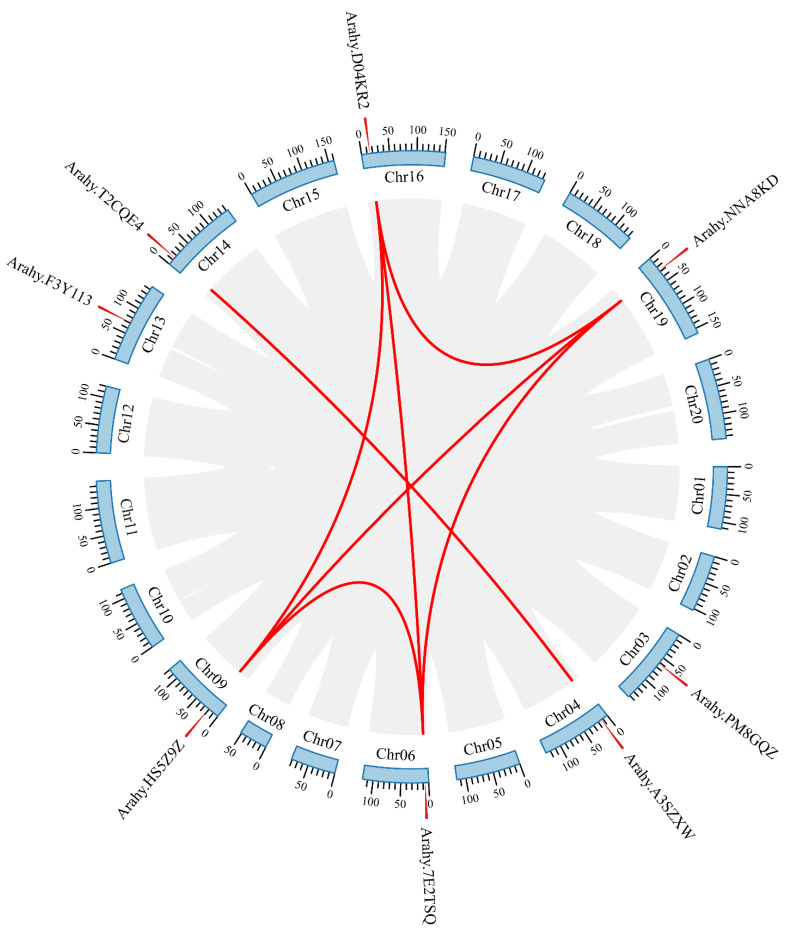
Chromosomal localization and intraspecific synteny of *AhPHY* genes. The blue areas indicate chromosomes, with the scale bar representing megabases (Mb). The gray areas indicate collinear regions, and the red lines indicate syntenic relationships between phytochrome genes of peanut.

**Figure 5 genes-14-01478-f005:**
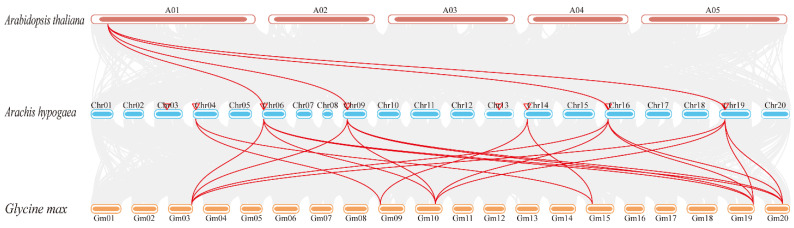
Interspecific synteny of *PHY* genes across different species. The red, blue, and yellow round rectangles indicate chromosomes of *Arabidopsis*, peanut, and soybean, respectively. The gray areas indicate collinear regions, and the red lines indicate syntenic relationships between phytochrome genes across different species.

**Figure 6 genes-14-01478-f006:**
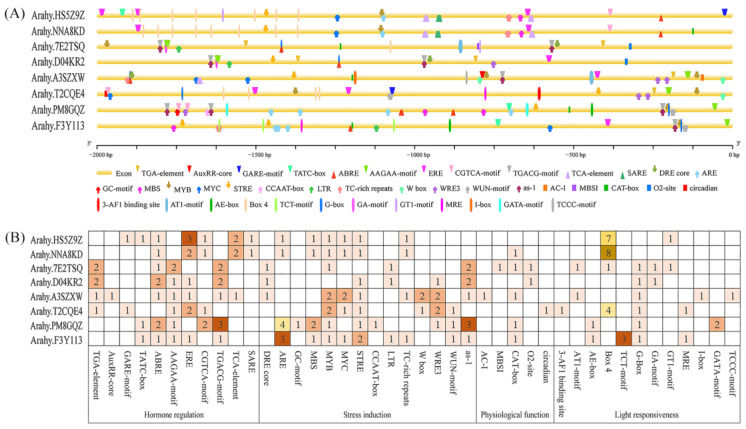
Prediction of *cis*-regulatory elements in the promoter regions of *AhPHY* genes. (**A**) Distribution diagram of *cis*-elements for *AhPHY* gene promoters, with diverse colored geometric figures representing distinct *cis*-elements; (**B**) The putative *cis*-elements were quantified and functionally classified based on their established roles in gene transcriptional regulation.

**Figure 7 genes-14-01478-f007:**
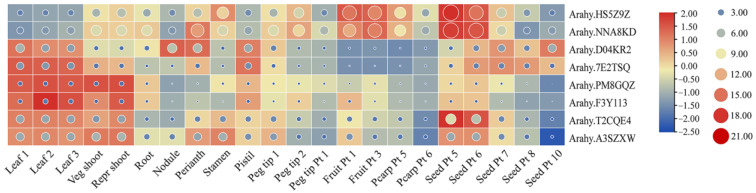
Expression profiling of *AhPHY* genes across various peanut tissues exhibited distinct spatiotemporal patterns. The square color scale of the heatmap indicates FPKM values following row normalization, with maximal and minimal expression denoted by red and blue, respectively. The circular color scale indicates raw FPKM values, where the surface area subtended by each circle was directly proportional to the expression level. Leaf 1 = lateral stem leaf; Leaf 2 = main stem leaf; Leaf 3 = seedling leaf; Veg shoot = vegetative shoot tip; Repr shoot = reproductive shoot tip; Root = root; Nodule = nodule; Perianth = perianth; Stamen = stamen; Pistil = pistils; Peg tip 1 = peg tip aerial; Peg tip 2 = peg tip below soil; Peg tip Pt 1 = peg tip to fruit Pattee 1; Fruit Pt 1/3 = fruit Pattee 1/3; Pearp Pt 5/6 = pericarp Pattee 5/6; Seed Pt 5/6/7/8/10 = seed Pattee 5/6/7/8/10.

**Table 1 genes-14-01478-t001:** Physicochemical properties and subcellular localization of *AhPHY* genes.

Gene ID	Amino Acid (aa)	Molecular Weight (Da)	Isoelectric Point (pI)	Instability Index (II)	Aliphatic Index	GRAVY	Subcellular Localization	Phosphorylation Site	
								Ser/Thr/Tyr	Total
*Arahy.HS5Z9Z*	1125	124,470.30	6.01	43.54	93.95	−0.081	Cyt, N	57/37/10	104
*Arahy.NNA8KD*	1113	123,208.83	5.80	45.13	95.22	−0.056	Cyt, N	58/36/9	103
*Arahy.7E2TSQ*	1125	124,620.66	6.05	43.27	95.07	−0.055	Cyt, N	60/36/8	104
*Arahy.D04KR2*	1125	124,612.76	6.14	43.40	95.50	−0.055	Cyt, N	61/35/8	104
*Arahy.A3SZXW*	1151	128,157.41	5.76	44.12	91.82	−0.171	Cyt, N	63/30/9	102
*Arahy.T2CQE4*	1147	127,672.95	5.80	43.90	91.72	−0.168	Cyt, N	64/28/9	101
*Arahy.PM8GQZ*	1101	122,424.19	5.80	45.79	91.04	−0.132	Cyt, N	62/24/10	96
*Arahy.F3Y113*	1116	123,960.97	5.72	46.13	91.32	−0.122	Cyt, N	61/26/11	98

Note: Cyt = cytoplasm; N = nucleus; Ser = serine; Thr = threonine; Tyr = tyrosine.

**Table 2 genes-14-01478-t002:** Secondary structure analysis of AhPHY proteins.

Gene ID	α-Helix (%)	Extended Strand (%)	β-Turn (%)	Random Coil (%)
*Arahy.HS5Z9Z*	48.00	14.31	5.42	32.27
*Arahy.NNA8KD*	48.61	14.56	5.30	31.54
*Arahy.7E2TSQ*	47.56	14.76	4.89	32.80
*Arahy.D04KR2*	48.36	14.76	5.69	31.20
*Arahy.A3SZXW*	47.35	14.68	5.30	32.67
*Arahy.T2CQE4*	47.69	14.73	5.58	32.00
*Arahy.PM8GQZ*	49.05	14.80	5.81	30.34
*Arahy.F3Y113*	49.28	14.70	5.29	30.73

## Data Availability

Data are contained within the article.

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
