# Peer review of "Genome-Wide Identification and Characterization of the Phytochrome Gene Family in Peanut"

_genes, 2023, doi:10.3390/genes14071478_

Round 1
Reviewer 1 Report
Authors present results that identify eight members of PHY gene family in peanut. Authors studied the gene structure, phylogenetics, physicochemical properties of proteins and their intracellular distribution. Moreover the genome distribution and syntene analysis was performed. Also results of transcriptomic studies describing the expression of presented genes in different peanut tissues were presented. Research is well planned and performed. Although it is based generally on available data imported from different databases and not supported by experimental data obtained by Authors, it represents still the important and new information characterizing members of PHY gene family in peanut.
In my opinion pictures presented in the research should be improved.
Fig. 2 In the description to Fig. 2 provide the aa sequences of motifs presented on Fig 2.
Fig. 6 It is hard to clearly define the cis-active element based only on the rectangle color. For example colors of AAGAA, MRE, GA and CAT motifs are to closely related (green). Authors could try to use not only colors, that are too closely related, but also different shapes, for example not only rectangles, but also squares, balls, triangles.
Related problem is in pairs MYB-SARE , DRE/G-box, TATC-TGA, MYC-ERE, Box4-TCA. Try to improve the unambiquity of cis-element description also in these pairs as described above.
Minor editing of English language required.
Reviewer 2 Report
The manuscript “Genome-wide Identification and Characterization of the Phytochrome Gene Family in Peanut” by Shen et al sent for publication to Genes deals with important topic such as characterization of in silico key gene family ( phytochrome) in peanut. The manuscript will be of interest to the scientific community working on the topic and could serve as a basic for further investigation in that crop. Below is the evaluation report:
Introduction:
The introduction is well written and point the main problems related to the topic.
Materials and methods:
This part needs improvement about expression pattern of the genes.
In section 3.6 of results, the authors described the expression pattern of AhPHY genes but this should be mentioned as well in the materials and methods.
Results:
This part is somehow well written.
Discussion:
This section is well written and supported by results.
Overall, the manuscript deserved to be published after addressing the comments and improve the Materials and Methods.
Reviewer 3 Report
Review for “Genome-wide Identification and Characterization of the Phytochrome Gene Family in Peanut”
This study attempt to identify and characterize the 8 candidate genes from phytochrome gene family in peanut. Several proporties of phytochrome genes were characterized by using various of analysis/assay approaches, including subcellular localization patterns, Phylogenetic relationship, cis-elements and functional relevance inferred from cis-elements, and spatiotemporal transcription pattern. By these analyses, this study attempt to enhance our understanding towards the expression pattern, function and evolution of PHY genes in peanut.
Major comments:
1. In section3.5, the putative functions of PHY genes were predicted by cis-regulatory element analysis. However, the predicted functions of these PHY genes should be further confirmed by other data types, e.g, co-expression network.
2. In section3.6, the expression patterns did not give any information about the functions of the PHY genes, which is important for our understanding of these genes.
3. For the overall manuscript, it is just a simple analysis of a PHY genes, which either did not give enough information of PHY genes or functional proporties of these genes.
Minor comments:
1. Section 2.1 should be “Prediction of xxx in peanut”. Data mining is trying to summarize information from big data. The analysis here was not looked like a data mining process.
2. From line 82 to line 98, the authors should provide parameter details of the tools during the analyses.
3. In line119, here it did not look like two Markov models for two motifs. The author should clarify how the queries had been conducted.
Word used could be improved
Round 2
Reviewer 3 Report
Although the study had conducted various of analyses, personally I was not convienced on the importance of the study. I still insist that further functional relative experiments like stress response/development patterns of these prediced PHYs are required for improving the depth of the analysis, which can be a strong hint on their roles in Penut. Download the public available RNA seq data from stress response/development profiling experiments and analyze the responsive patterns and functional relevance of the PHYs could be a way for the improvement.
Word usage should be further checked for preciously interpret and describe the results.
